# ALTERNATING ROLES DIALOG MODEL WITH LARGE-SCALE PRE-TRAINED LANGUAGE MODELS

## ABSTRACT

Existing dialog system models require extensive human annotations and are difficult to generalize to different tasks. The recent success of large pre-trained language models such as BERT and GPT-2 (Devlin et al., 2019; Radford et al., 2019) have suggested the effectiveness of incorporating language priors in down-stream NLP tasks. However, how much pre-trained language models can help dialog response generation is still under exploration. In this paper, we propose a simple, general, and effective framework: Alternating Roles Dialog Model (ARDM). ARDM models each speaker separately and takes advantage of the large pre-trained language model. It requires no supervision from human annotations such as belief states or dialog acts to achieve effective conversations. ARDM outperforms or is on par with state-of-the-art methods on two popular task-oriented dialog datasets: CamRest676 and MultiWOZ. Moreover, we can generalize ARDM to more challenging, non-collaborative tasks such as persuasion. In persuasion tasks, ARDM is capable of generating human-like responses to persuade people to donate to a charity.

## 1 INTRODUCTION

It has been a long-standing ambition for artificial intelligence researchers to create an intelligent conversational agent that can generate human-like responses. Recently data-driven dialog models are more and more popular. However, most current state-of-the-art approaches still rely heavily on extensive annotations such as belief states and dialog acts (Lei et al., 2018). However, dialog content can vary considerably in different dialog tasks. Having a different intent or dialog act annotation scheme for each task is costly. For some tasks, it is even impossible, such as open-domain social chat. Thus, it is difficult to utilize these methods on challenging dialog tasks, such as persuasion and negotiation, where dialog states and acts are difficult to annotate.

Eric & Manning (2017) proposed a simple sequence-to-sequence architecture that requires no explicit annotations. The model learns to extract information from dialog history with attention and copy mechanism. However, due to the limited language modeling capabilities in the previous model, Sequicity (Lei et al., 2018), which reuses belief states as inputs for supervision, outperforms Eric & Manning (2017)'s method significantly in recent dialog datasets. But with the success of large pre-trained language models such as BERT (Devlin et al., 2019) and GPT-2 (Radford et al., 2019), we re-examine Eric & Manning (2017)'s method and investigate how large-scale pre-trained language models can help dialog tasks.

Previous large-scale pre-trained language models are used to tackle documents with only one narrator. However, in dialogs, two speakers have different roles; therefore, their language model distributions are very different from each other. For example, customer service agents speak very differently to their customers. To address this issue, we propose ARDM, a dialog model that encodes and decodes different speaker utterances in alternating order with two pre-trained large-scale language models. To investigate whether ARDM can help dialog response generation, we evaluate its performance on three different task-oriented dialog datasets: CamRes676, MultiWOZ, and PersuasionForGood . The first two datasets are traditional information request dialog datasets with well-defined automatic evaluation metrics on task completion. By contrast, PersuasionForGood is a new dataset that focuses on persuading people to donate to a charity. There is no explicit dialog state defined in this task as such non-collaborative dialogs have various dialog actions.

We observe that ARDM is capable of improving task-oriented dialog tasks performance over the previous state-of-the-art methods without incorporating any explicit supervision from belief states or dialog acts. Also, due to ARDM's simplicity and generality, one can rapidly build a dialog prototype on different types of applications using only conversations without any manual annotations. We also found that ARDM works well on complex dialogs, such as persuasion. The model generates dialog responses that successfully persuade people to donate to a charity, suggesting the potential of ARDM being used in wide-scale real-world settings.

## 2   RELATED WORK

Traditional dialog systems consist of a dialog manager to maintain dialog states and control the conversation flow. However, a dialog manager requires extensive manual annotations for training the sub-modules such as dialog state tracker or policy decision-maker. An alternative is to model dialog without explicitly modeling belief states. Specifically, Eric & Manning (2017) proposed a recurrent neural dialogue architecture using a sequence-to-sequence model that utilizes copy-mechanism to copy history information directly from raw dialog history. This method achieved the state-of-the-art results on DSTC2 (Henderson et al., 2014), which is a simple dialog restaurant booking task with abundant data. However, such method did not perform well on more complex dialog task data sets CamRes676 (Wen et al., 2017) and KVRET (Eric et al., 2017). Sequicity (Lei et al., 2018) attributed the bad performance of Eric & Manning (2017)'s method to the omission of belief tracker. They introduced the concept of belief span and added belief tracker back to the model and achieved state-of-the-art performance.

Compared to Sequicity, Eric & Manning (2017)'s method provides a more general framework that reduces manual dialog state, user intent, and dialog act labeling by bypassing any symbolic annotations. Such a model can apply to datasets with no or partial annotations of belief states. In a real-world setting, if the dialog task introduces new slot values in belief states (i.e. a new type of food), Sequicity will suffer from the belief span decoder error in response generation. Thus, Eric & Manning (2017)'s method may be potentially more robust than Sequicity in this situation. Besides, if the task requires belief states for database search, we can treat belief tracking as a separate task. We can train a good belief tracking with only a small amount of annotated data, which reduces the annotation required and it is easier to fix errors. Also, since belief states are a set of important entities condensed from dialog history (i.e., often exact words from utterances), they do not introduce extra information to the model. Therefore, a dialog model with powerful representation learning should learn a form of belief states information automatically without human annotations as the scaffold.

Recent success of BERT (Devlin et al., 2019) and GPT2 (Radford et al., 2019) suggests the possibility of using large pre-trained language models to enhance Eric & Manning (2017)'s method. There are some studies of applying large pre-trained language model to dialog generation. Transfer-Transfo (Wolf et al., 2019) fine-tuned the pre-trained language model GPT (Radford et al., 2018) on Persona-Chat dataset (Zhang et al., 2018) and obtained significant improvements on chitchat dialog generation, suggesting the potential of fine-tuning large pre-trained language model on other dialog response generation tasks. A more recent work (Budzianowski & Vulic, 2019) adopted the framework of TransferTransfo and made the first attempt to leverage large pre-trained language models GPT and GPT-2 on task-oriented dialog generation, but it included belief states modeling as the input and did not achieve better results than the baseline. We propose to model dialogs without any annotation but rely on pre-trained large scale language models that alternate.

Previous work shows that modeling speaker roles in conversation is beneficial for language understanding (Chi et al., 2017; Chen et al., 2017; Su et al., 2018). Other researchers model persona information to generate language with different speaking styles (Li et al., 2016; Joshi et al., 2017). Zhao & Kawahara (2019) propose a relative speaker modeling method, where only the relative role instead of the absolute identity of the speaker is modeled. Our method is similar to Zhao & Kawahara (2019) in the spirit of modeling relative speaker relationship, but we focus on learning role-specific language models through utterances from different speakers, instead of explicitly taking role embeddings as input.

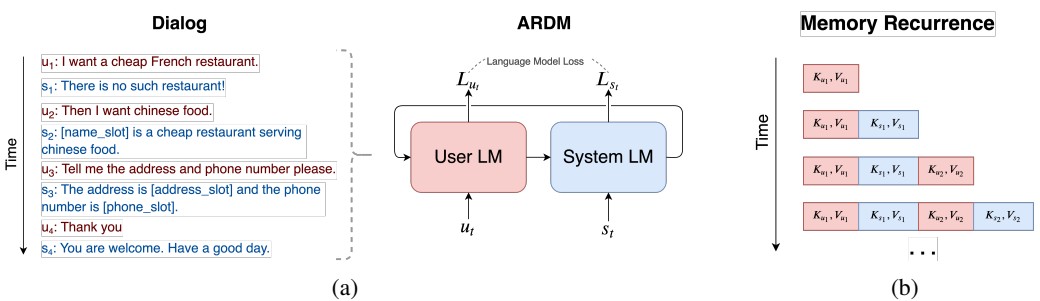

(a)                                        (b)

Figure 1: Alternating Roles Dialog Model (ARDM) Overview. (a) shows how we feed the entire dialog to ARDM. (b) shows the recurrence mechanism we used to preserve memory.

## 3  APPROACH

Our goal is to leverage large pre-trained language models to improve dialog response generation. Favoring Eric & Manning (2017)'s approach without using additional annotations such dialog states or dialog acts, we propose Alternating Roles Dialog Model (ARDM) by compositing two separate pre-trained language model in alternate order to learn the user and system utterance distribution. Figure 1 shows an overview of ARDM.

### 3.1  ALTERNATING ROLES DIALOG MODEL

We aim to model both user and system utterances distribution simultaneously. Given a multi-turn dialog ($d$) between a user ($u$) and a system ($s$), we can represent $d$ as a series of utterances $\{u_1, s_1, u_2, s_2, \ldots, u_T, s_T\}$, where $T$ denotes the total number of turns. We decompose the probability distributions over the utterances in $d$ into two language models for the user and system respectively, denoted as $p_u$ and $p_s$. Then we define a dialog model $p(d)$ with the equation:

$$p(d) = \prod_{t=1}^{T} p_u(u_t|u_{<t}, s_{<t})\, p_s(s_t|u_{\leq t}, s_{<t}) \tag{1}$$

$p_u$ and $p_s$ are standard language models where the task is to predict the next token given the preceding context. For an utterance $u_t$ or $s_t$ with $m$ tokens $\{w_1, \ldots, w_m\}$, the joint probability of an utterance is as follows:

$$p_u(u_t|u_{<t}, s_{<t}) = \prod_{i=1}^{m_{u_t}} P(w_i|w_{<i}, u_{<t}, s_{<t}) \tag{2}$$

$$p_s(s_t|u_{\leq t}, s_{<t}) = \prod_{i=1}^{m_{s_t}} P(w_i|w_{<i}, u_{\leq t}, s_{<t}) \tag{3}$$

Finally, we train the dialog model by maximizing the likelihood over Equation 1.

We apply a simple memory mechanism to grant the model the capability of memorizing conversation history. For an utterance at turn $t$, we reuse the hidden states $h_{\leq t-1}$ stored in the memory $M_{t-1}$ to obtain $h_t$, and store the $h_t$ back to the memory as $M_t$. As for the pre-trained Transformer language model, we implement the memory mechanism using self-attention given the query/key/value features denoted as $Q, K, V$, where the equation is defined as:

$$\text{Attention}(Q, K, V) = \text{softmax}(QK^TV) \tag{4}$$

For simplicity, we assume there is only one layer in Transformer, and $h_t$ is the hidden states which consist of $N$ vectors for the current input $N$ tokens in the utterance at time $t$. Then a recurrence relation for $h_t$ is defined by computing $Q_t$, $K_{\leq t}$, $V_{\leq t}$ from $h_{\leq t-1}$ and the current utterance. In

practice, we reuse $K_{\leq t-1}$ and $V_{\leq t-1}$ (i.e. history keys and values) as $M_{t-1}$ instead of $h_{t-1}$ to avoid recomputing history information. Therefore, the final $h_t$ is computed as:

$$M_{t-1} = [K_{\leq t-1}, V_{\leq t-1}] \tag{5}$$
$$K_{\leq t}, V_{\leq t} = [K_{\leq t-1}; K_t], [V_{\leq t-1}; V_t] \tag{6}$$
$$h_t = \text{Attention}(Q_t, K_{\leq t}, V_{\leq t}) \tag{7}$$

One can use $h_t$ (consisting of vectors for each token) to get each token's probability to calculate the language model cross entropy loss to maximize $p(w_i | w < i, u_{<t}, s_{<t})$, shown in Figure 1.

However, one major drawback of this memory mechanism is that the memory consumption grows as the number of turns increases, until a point that the dialog cannot continue because of the memory limit. A straightforward way to solve this is to discard the distant history. But because most dialogs lengths in our datasets can fit in the GPU memory limit (i.e., approx. 1,000 tokens for 11GB GPU), we leave the memory issue for future work.

## 3.2 TRAINING DETAILS

We initialize the user and the system language model with a large pre-trained language model GPT-2 small with 117M parameters (Radford et al., 2019). It is a Transformer (Vaswani et al., 2017) model with 12 heads, 768 hidden size, and 12 layers. The model is trained on a large scale corpus called WebText extracted from Reddit with at least three upvotes. The tokenizer is 50,257 size byte pair encoding (BPE) (Sennrich et al., 2016) that can encode and decode any text in a lossless manner to avoid out-of-vocabulary tokens. We follow a special format in GPT-2 as the "trigger" so that the model can zero-shot dialog response generation, by prefixing the user role token "A:" or "B:", and suffixing the end of utterance token "\n\n\n". This "trigger" approach is similar in other zero-shot scenarios mentioned in GPT-2 paper (e.g., that a "TL;DR" token can trigger GPT-2 to summarize the input text.) We further fine-tune ARDM on the specific task dataset. We apply AdamW optimizer (Loshchilov & Hutter, 2019), and the number of warm-up steps is set to be the number of batches in one epoch. The learning rate is set to $3 \times 10^{-5}$, and the dropout rate is set to $0.1$ for all tasks.

## 3.3 DECODING DETAILS

We decode utterances by nucleus sampling (Holtzman et al., 2019) with different hyper-parameters (top-p, top-k) for down-stream dialog tasks. We also vary the temperature of $T < 1$ to find the best setting for the specific down-stream dialog task. To handle both situations in the evaluation and the real-world use case, we have two decoding modes. For evaluation mode, we feed all past ground truth history before turn $t$ to generate the corresponding utterance, so that we can evaluate the quality of generated dialog responses without concerning about the conversion flow. While in a real-world use case, we do not have ground truth history, and therefore we use the memory from previously generated responses and let the model dynamically interact with a human or another bot in turns. Because dialogs have different lengths, it is hard for ARDM to efficiently decode responses using traditional batch padding method. As a solution, we develop a dynamic dialog filtering algorithm to support fast decoding in batch. Such method speeds up the generation eight times faster. Please refer to Appendix B for the method's details.

## 4 EXPERIMENTS AND RESULTS

Data scarcity is one of the biggest challenges in dialog research. It is costly to collect human-human conversations under a specific setting. It is even more time-consuming to annotate belief states and dialog acts. With the success of transfer learning in NLP, we aim to mitigate the low-resource problem with the large pre-trained language model. We validate our proposed ARDM on three task-oriented dialog datasets, CamRest676, MulitWOZ, and PersuasionForGood.

## 4.1 CAMREST676

CamRest676 is a relatively small dataset with 408/136/136 dialogs for train/validation/test. We follow Sequicity (Lei et al., 2018) to delexicalize tokens such as restaurant names, phone numbers, postcodes by replacing them with their slot names in utterances. We prepend database search results

to the system utterance. An example database search results are "restaurant;3", where the first slot indicates its dialog domain, which is always "restaurant" in CamRest767, and the second slot represents the number of matched items in the database. We use nucleus sampling for all methods in decoding for a fair comparison. Here, we set top-p $0.2$ and temperature $0.7$ for our model. We use **BLEU-4** and **Success F1** to evaluate language generation quality and Success F1 to evaluate task success. Success F1 computes the F1 score of the generated responses on requested slots such as an address, phone number, or food type. Other than Sequicity, we also compare results by using GPT-2 alone as a language model for the entire dialog.

### 4.1.1 RESULTS

We first test our method on a restaurant search dataset, CamRest676 (Wen et al., 2017).

| Model | Entity Match rate | Ground Truth Belief State | | Generated Belief State | |
|---|---|---|---|---|---|
| | | BLEU-4 | Success. F1 | BLEU-4 | Success. F1 |
| Regular Expression | 0.960 | - | - | - | - |
| Sequicity | 0.923 | 21.4 | 0.852 | 21.4 | 0.853 |
| Sequicity (w/o RL) | 0.940 | 22.9 | 0.821 | 23.4 | 0.834 |
| GPT-2-finetune | - | 21.8 | 0.851 | 19.2 | 0.862 |
| ARDM | - | **26.0** | **0.875** | **25.2** | **0.871** |
| ARDM (50% data) | - | 25.9 | 0.859 | 23.4 | 0.851 |

Table 1: Results on CamRest676 dataset.

Table 1 shows all models' results with ground truth belief state or generated belief state. We first use ground truth belief state in all methods to evaluate their response generation quality. ARDM achieves the best BLEU and Success F1 score. We observe that after fine-tuning GPT-2 on the CamRest676, it achieves similar results compared to the previous state-of-the-art method, Sequicity with reinforcement fine-tuning. This suggests pre-trained large-scale language model, such as GPT-2, transfers the meaningful representations to help fine-tuning. However, without the alternating mechanism, GPT-2 alone does not perform as well as ARDM in terms of both BLEU-4 and Success F1, especially in BLEU-4 (improved 19%). Without modeling the speaker role, the model blends two speakers language distribution and ignores the inherent speaker role difference. Moreover, to test if our model preserves its performance with even less training data, we reduce the training data to 50%, and the performance only drops slightly. With half of the training data, our method still performs significantly better than Sequicity. This result suggests ARDM is robust on low-resource settings due to the advantage of the large-scale pre-training language model.

We also evaluate all models with generated belief states instead of ground truth belief states. Sequicity generates belief tracker results, and its Entity Match rate is 0.927. Our model does not have a state tracker, so we write a separate simple regular expression to extract the occurrence of entities that appear in the database to support our model. Such state tracker achieves 0.960 in Entity Match rate. It suggests that state tracking may be accomplished in more straightforward ways other than training a neural network model on a large set of annotated data. With a simple state tracker, our proposed method still performs better than Sequicity, which trains the belief state and the response generation task jointly.

### 4.2 MULTIWOZ

Here, we only use the ground truth database search result to be consistent with other methods. We perform delexicalization which is mentioned in the original MultiWOZ (Budzianowski et al., 2018). We prepend the database search results to the system response for as conditional input. Also, the database results now contain information about whether the booking is successful or not (i.e., succeed or fail). Note that we do not use belief state or dialog act annotation provided by the dataset to train ARDM. We set the top-p to $0.2$ and the temperature to $0.7$. The results are evaluated on BLEU-4, Inform Rate, and Success Rate. Inform and Success Rate measure whether the system response provides the recommendations and requested information given in the goal. We compare our model to the attention-based seq2seq model which is proposed as the MultiWOZ Baseline (Budzianowski

et al., 2018), the HDSA (Chen et al., 2019) model that incorporates dialog act supervision as an inductive prior for model architecture, and the LaRL (Zhao et al., 2019) model which leverages latent action modeling and reinforcement learning to improve performance. We do not compare with GPT-2-finetune with our model in MultiWOZ because GPT-2-finetune's performance on CamRest676 is significantly worse than our model. We normalize the time's slot value in all dialogs into the 24-hour format and perform tokenization via spaCy[1]. We found that different papers report results with different versions of the evaluator, which makes it difficult to compare different methods fairly. We explain the differences among all versions of the evaluator in Appendix A. In this paper, we follow LaRL's evaluator implementation, as it is more reasonable than others. We re-evaluate results for all methods with the same evaluator to ensure fairness.

### 4.2.1 RESULTS

| Model | Supervision | | Inform (%) | Success (%) | BLEU-4 |
| | Dialog State | Dialog Act | | | |
| --- | --- | --- | --- | --- | --- |
| Human | - | - | 98.9 | 96.5 | - |
| Baseline | ✓ | ✗ | 82.5 | 72.9 | 18.9 |
| HDSA | ✓ | ✓ | **87.7** | 73.4 | **23.6** |
| LaRL | ✓ | ✗ | 82.8 | **79.2** | 12.8 |
| ARDM | ✗ | ✗ | 87.4 | 72.8 | 20.6 |

Table 2: Results on MultiWOZ. Supervision denotes whether a model leverages dialog state or/and dialog act annotations. All models use the ground truth dialog state for database search. ARDM without supervision from annotation can still achieve comparable results.

The evaluation results are shown in Table 2. Without any supervision from dialog states or dialog acts, ARDM significantly outperforms the MultiWOZ Baseline and LaRL on BLEU-4 and Inform rate, and is on par with HDSA. However, HDSA uses dialog act supervision and a large pretrained language model, BERT. Our model requires no annotation and can achieve similar results. This suggests our speaker role modeling and large-scale pre-training methods work similarly as the useful dialog act annotations. All the results show that our method's excellent performance remains consistent in multi-domain dialogs.

We analyze the generated responses and find that if multiple domains have appeared in the conversation history, our model tends to make mistakes in answering the right domain for user requests. This finding suggests that the Maximum Likelihood Estimation (MLE) has limitations in directly optimizing the metric, while reinforcement Learning (RL) can hugely improve the task completion in a dialog system. This is why LaRL has a higher Success rate. However, we also observe that LaRL has a low BLEU-4 score, which indicates low readability in responses. Therefore, there is a trade-off between the generation quality and the task success rate in the RL setting.

### 4.3 PERSUASIONFORGOOD

To showcase ARDM's performance on a dialog dataset where it is much more difficult to obtain belief states and dialog act annotations, we train and evaluate our model on PersuasionForGood (Wang et al., 2019) dataset. In this dataset, the persuader must persuade an assigned persuadee (i.e., a person who is asked to donate) to donate money (from their task payment) to a charity called "Save the Children". This dataset has a much larger vocabulary size (8,141) than the previous task-oriented dialog datasets due to its non-collaborative dialog property. The conversation content is richer because two speakers are negotiating back and forth. The dataset consists of 1,017 dialogs where only 300 dialogs are annotated with dialog acts. Therefore, models that require dialog state or dialog act annotation are not applicable in this dataset. ARDM does not require dialog acts for training. As ARDM has already learned good language priors through pre-training, which makes it applicable on PersuasionForGood. Due to the similarity to chit-chat, we use TransferTransfo (Wolf et al., 2019) model as a strong baseline model. TransferTransfo is also based on large pre-trained

---

[1]https://spacy.io/

language model, but it uses token type embedding to encode role information of the speaker. We concatenate all history text together as input to generate the next utterance.

To generate diverse responses, we decode the response using the nucleus sampling (Holtzman et al., 2019) with a top-p of 0.9 and a temperature of 0.7. It is impossible to conduct an automatic evaluation on task success on this task due to the lack of annotation. We use perplexity, BLEU-1, and BLEU-2 to evaluate each model's language generation quality. We also conduct a human evaluation to validate each model's task success rate. We show some generated examples in the Appendix to provide more information on both models' generation quality.

### 4.3.1 RESULTS

| | Perplexity ↓ | BLEU-1 ↑ | BLEU-2 ↑ | Human Preference ↑ | Average Donation Amount ↑ |
|---|---|---|---|---|---|
| TransferTransfo | 19.9 | **17.0** | **8.57** | 34.7% | 0.538 |
| ARDM | **10.1** | 16.5 | 6.44 | **65.3%** | **0.807** |

Table 3: Automatic Evaluation and Human Evaluation Results

| Role | Utterance |
|---|---|
| SYS | Hi, how are you doing today? |
| USR | I am good! |
| SYS | Have you ever heard of the charity Save the Children? |
| USR | No, I have not heard about it. Would you like to talk about it? |
| SYS | Save the Children is an international non-governmental organization that promotes children's rights, provides relief and helps support children in developing countries. |
| USR | That's cool. It is an organization for the good of this world. |
| SYS | Yes, it is. In the first two months of 2018 alone, 1,000 children were reportedly killed or injured in intensifying violence in Syria. |
| USR | That's sad. |
| SYS | We do need to do something about that. One of the things that we can do is make sure that we have a system in place that will ensure that every penny we spend goes directly to the children in need. |
| USR | I agree. |
| SYS | How much do you like to donate to the charity now? Your donation will be directly deducted from your task payment. You can choose any amount from $0 to all your payment ($2). |
| USR | I am willing to donate full my payment $2. |
| SYS | Wow, that is awesome! That is so kind of you! |
| USR | You are welcome. |

Table 4: A conversation between a machine persuader trained by ARDM and a human persuadee.

Table 3 shows the results for PersuasionForGood. Because ARDM applies better speaker modeling and recurrence mechanism, our model achieves lower perplexity compared to TransferTransfo. In terms of BLEU scores, TransferTransfo is better than ARDM. However, BLEU-1 cannot reflect the actual generation quality because a random sentence with common tokens the, of, is, are already has 10.0+ BLEU-1 score. Also because the validation set only contains 100 samples, the result can have a high variance. To comprehensively evaluate each model's performance, we recruit 14 human evaluators to chat with the two persuasive systems ten times to avoid the randomness produced by each model. In total, we collected 140 ratings. We ask them to select a preferred chat-bot and indicate how much they are willing to donate after talking to the chat-bot. As a result, human judges prefer ARDM over TransferTransfo and tends to donate more when talking to ARDM produced chat-bot. Our model achieved 27% more donations compared to TransferTransfo. This indicates that our systems are more persuasive. In some examples, such as the one in Table 4, our model generates coherent, natural, and persuasive responses.

## 5 ERROR ANALYSIS

Since CamRest676 is similar to MultiWOZ in terms of task content and dialog structure, we only describe the errors in MultiWOZ for simplicity. We randomly selected 30 generated error responses

from our model with zero inform and success score. To our surprise, we observed that nearly 63.3% of errors are not really mistakes. It is mainly due to the limitation of the automatic evaluator. For example, at turn one, the user asks about a restaurant, and the ground truth system response is "the [restaurant_name] is located at . . . ", but the generated system response is "what food preference do you have?". Our generated response is correct with respect to the dialog context. It is narrowing down the restaurant choices before providing a restaurant recommendation. However, the evaluator sticks to the only possible response it has. Unless the user can dynamically interact with the system, there is no good way to change such mistakes in the automatic evaluator. We find that another 20% errors our model makes are when the system asks information the user already provided. This type of errors calls for a better history representation. Another 10% errors are due to ignoring the user's request for information, such as phone number. However, when we look at the ground truth responses, some crowd workers also made such errors. So resolving these errors requires a cleaner training dataset. Finally, the rest of 6.7% errors are about incorrect dialog domain understanding. For example, the user is asking for a hotel, but we present a restaurant recommendation. This is because of the data noise during the delexicalization process in which some domain labels are wrong.

The donation persuasion system trained with TransferTransfo and our model has some common problems, such as inconsistency, lack of logic, and hallucination. For example, if the persuader provides the information about "Save the Children", then the persuadee asks "Can you tell me more about it?". The system ends up providing the same information as before. It also sometimes makes up facts that have never happened, such as "Save the Children has an operation about a hurricane in Hawaii". All those errors would prevent users from trusting the bot, and therefore resulting in less donation. However, we also observe that users have a higher tolerance for errors in the persuasion setting than the customer service setting. Overall, our model performs better on PersuasionForGood by having longer and diversified utterances. This suggests our model which utilizes alternating parameters for different speaker roles is more effective than TransferTransfo which only injects role information into the input embedding.

## 6 DISCUSSIONS AND ETHICAL CONSIDERATION

ARDM models speakers separately on top of a large pre-trained language model. Such simple adaptation demonstrates substantial performance gain. We suspect it is because the interleaved structure of two language models provides a collaborative learning frame of both the user and the system language distribution modeling. The memory is the only way for the user and system to communicate, as they do not share any weights in their networks. Thus, the user encoder needs to learn useful representations to make the system model for understanding its intent. Similarly, the system needs to do the same for the user model to improve its understanding. This alternative repeating process forces both the user and system models to preserve the dialog history effectively in the memory. One can interpret the memory as the implicit representation of belief states or dialog acts.

Another benefit of ARDM is that we will obtain both user and system utterance generators. We can let the two models talk to each other to generate new self-play dialogs (Silver et al., 2017). We show some self-play dialog examples in the Appendix E. With self-play, one can rapidly build a large scale dialog dataset using adversarial filtering (Zellers et al., 2018). Such models can be used in reinforcement learning as user simulator to study complex dialog strategies as well.

Persuasion is a double-edged sword. Given the fast development of dialog systems, an ethical design principle must be in place throughout all stages of the development and evaluation. We choose the donation task is because it is a relatively simple task that benefits children. Second, when deploying the persuasive agents in real conversations, we need to keep the users informed of the nature of the system. By revealing the identity of the persuasive agent, the user should also have options to communicate directly with the human team behind the system. Lastly, by investigating persuasive dialog systems, we also envision to use them as an educational tool for the general public to learn to defend themselves against machine persuasion.

## 7 CONCLUSIONS

We propose to build Alternating Roles Dialog Model (ARDM), a simple, general, and effective dialog method that models user and system separately with large-scale pre-trained language models. Since ARDM does not require any annotations, it generalizes to different dialog applications. Experimental results on CamRest676 and MultiWOZ suggest that ARDM outperforms or on-par with the current state-of-the-art methods that use manual annotation information, such as belief states and dialog acts. Furthermore, we find our model's excellent performance generalizes to more complex non-collaborative dialog settings. It can generate high-quality responses to persuade people to donate to charity. However, the easiness of training ARDM raises concerns about the misuse of the model in scenarios such as sales, harassment, or scam on a mass scale. We caution the public in deploying such systems in the real world.

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

## A  MULTIWOZ EVALUATOR INCONSISTENCY

We rerun baseline models to compare our methods and find discrepancy among different papers' reported results. In order to understand the reason, we compared between LaRL's evaluator [2] and MultiWOZ Baseline's evaluator [3]. We found that they make different assumptions to handle the "train" domain (line 637-639 at LaRL evaluator.py). After carefully analyzing the code and discussing with authors of these two papers, we believe that LaRL's evaluator is more reasonable. However, in LaRL, the authors reported MultiWOZ Baseline's scores with a different evaluator. Therefore, we re-evaluated all methods, including LaRl, HDSA, and MultiWOZ Baseline using the same evaluator for fairness.

|  | Baseline Evaluator | | LaRL Evaluator | |
| --- | --- | --- | --- | --- |
|  | Inform | Success | Inform | Success |
| Human | 75.7% | 67.9% | 90.0% | 82.3% |
| Human (the cleaned version) | 82.4% | 78.9% | 98.9% | 96.5% |
| MultiWOZ Baseline | 71.3% | 61.0% | 82.5% | 72.9% |

Table 5: Re-evaluation Results on MultiWOZ.

## B  DYNAMIC DIALOG FILTERING ALGORITHM

Since ARDM takes the entire dialog as input, so different input has different number of turns and different length in each turn. Also, there are two sub-language models in ARDM. If we pad the utterance in each turn and concatenate results of all turns, it is inefficient. Instead, we introduce a filtering algorithm to dynamically release memory once a dialog is finished. This enables us to speed up our model four times in training and eight times in inference. Please refer to our code for details.

---
**Algorithm 1** Dialog Filtering algorithm

---
Batch Size = $n$
Initialize Memory $M$ for $n$ workers.
**for** Batch = $1, 2, 3, \ldots$ **do**
    $T$ = maximum number of turns in Batch
    **for** Turn = $1, 2, 3, \ldots, T$ in Batch **do**
        Filter $M$ and Batch for the finished worker
        Feed remaining Batch to the model
        Compute loss and gradients
        $\theta_{old} \leftarrow \theta$
    **end for**
**end for**

---

[2] https://github.com/snakeztc/NeuralDialog-LaRL/blob/master/latent_dialog/evaluators.py
[3] https://github.com/budzianowski/multiwoz/blob/master/model/evaluator.py

## C  TRANSFERTRANSFO EXAMPLES

| Role | Utterance |
|------|-----------|
| ER | hello! |
| EE | Hello |
| ER | have you heard of the charity save the children? |
| EE | no, I haven't heard it |
| ER | it's an international non - governmental organization that promotes children's rights, provides relief and helps support children in developing countries. |
| EE | ok |
| ER | so would you like to donate some of your payment to this charity? |
| EE | can you tell me more information about it? |
| ER | save the children is an international non - governmental organization that promotes children's rights, provides relief and helps support children in developing countries. |
| EE | which developing countries? |
| ER | the united states and latin america. have you heard of this charity? |
| EE | no, I haven't |
| ER | it's an organization that helps ensure children's rights to health, education and safety. |
| EE | sounds cool |
| ER | have you ever donated to this organization? |
| EE | no |

Table 6: A conversation between a machine persuader (ER) trained by TransferTransfo and a human persuadee (EE)

| Role | Utterance |
|------|-----------|
| ER | hello! |
| EE | hi |
| ER | how are you today? |
| EE | I am good |
| ER | have you ever heard of save the children? |
| EE | no, I haven't |
| ER | it is an organization that helps children around the world. it's a great organization that does good work and is highly rated with many positive rewards. |
| EE | good |
| ER | have you ever heard of it? |
| EE | no, I haven't heard of it |
| ER | it's a great organization that does good work and is a huge part of the world's population. |
| EE | okay |
| ER | so, are you interested in donating? |
| EE | yes, I am |
| ER | how much would you like to donate to the charity? |
| EE | $1 |
| ER | wow, that is really great.  that is really great to hear.  so, are you going to donate anything at this time?? |

Table 7: A conversation between a machine persuader (ER) trained by TransferTransfo and a human persuadee (EE)

## D  PERSUASIONFORGOOD HUMAN-MACHINE EXAMPLES

We show some more examples of dialogs generated using our proposed ARDM.

| Role | Utterance |
|------|-----------|
| ER | Hello! How are you doing today? |
| EE | I am doing well! |
| ER | Awesome. Have you heard about the Save the Children charity? |
| EE | Yes I have heard about it. |
| ER | Save the Children is an international non-governmental organization that promotes children's rights, provides relief and helps support children in developing countries. |
| EE | So which developing countries do they help? |
| ER | Save the Children is working on supporting children in the US and all over the world. |
| EE | that's cool. I want to help children as well. |
| ER | That's awesome! I know there are a lot of kids in the world that are in dire situations. |
| EE | Right. What I should I do to help them? |
| ER | Save the Children is a great organization, so you should donate some of your payment to them. |
| EE | how much is my task payment? |
| ER | I am not sure exactly how much you will donate, but you can donate any amount from $0 to all of your payment ($2). |
| EE | I would like to donate $1.00. |
| ER | That's awesome. Your donation will make a huge impact for those kids. |
| EE | Thank you. |

Table 8: A conversation between a machine persuader (ER) trained by ARDM and a human persuadee (EE)

| Role | Utterance |
|------|-----------|
| ER | Hi, how are you doing today? |
| EE | I am good! |
| ER | Have you ever heard of the charity Save the Children? |
| EE | No, I have not heard about it. Would you like to talk about it? |
| ER | Save the Children is an international non-governmental organization that promotes children's rights, provides relief and helps support children in developing countries. |
| EE | That's cool. It is a organization for the good of this world. |
| ER | Yes, it is. In the first two months of 2018 alone, 1,000 children were reportedly killed or injured in intensifying violence in Syria. |
| EE | That's sad. |
| ER | We do need to do something about that. One of the things that we can do is make sure that we have a system in place that will ensure that every penny we spend goes directly to the children in need. |
| EE | I agree. |
| ER | How much do you like to donate to the charity now? Your donation will be directly deducted from your task payment. You can choose any amount from $0 to all your payment ($2). |
| EE | I am willing to donate full my payment $2. |
| ER | Wow, that is awesome! That is so kind of you! |
| EE | You are welcome. |

Table 9: A conversation between a machine persuader (ER) trained by ARDM and a human persuadee (EE)

| Role | Utterance |
|------|-----------|
| ER | Hello |
| EE | Hi |
| ER | How are you doing today? |
| EE | I am good. thank you |
| ER | I was wondering if you would be interested in donating a portion of your payment from this HIT to a children's charity called Save the Children? |
| EE | Of course, I would like to. Can I know their website address? |
| ER | Yes, URL |
| EE | Oh, cool. They are a good organization. |
| ER | How much would you like to donate? |
| EE | I would like to donate $1.00 |
| ER | That is great! The research team will collect all donations and send it to Save the Children. |
| EE | You are welcome. |

Table 10: A conversation between a machine persuader (ER) trained by ARDM and a human persuadee (EE)

# E    PERSUASIONFORGOOD SELF-PLAY EXAMPLES

We also present several self-play examples generated using our proposed ARDM.

| Role | Utterance |
|------|-----------|
| ER | Hello! How are you doing today? |
| EE | Good, thank you. How are you? |
| ER | I am good, thank you. Have you heard of the charity Save the Children? |
| EE | Yes, I have heard about it. I would like to donate a dollar to it. |
| ER | That is wonderful, it is so important to provide education and support to the children. I am sure you will agree that this is a great cause. Do you donate to charities? |
| EE | Yes, I do donate to several charities. |
| ER | Are you sure you want to donate a dollar to Save the Children? Children all over the world are suffering due to poverty, war, and other issues. Would you consider donating a little bit of your incoming task payment? |
| EE | I may donate a dollar. I don't have a lot of money right now but I would like to help. |
| ER | That is great to hear. I know your donation is a small amount but the impact this will have on the world can be significant. Your donation would be directly deducted from your task payment. |
| EE | That is true. I will donate one dollar to the charity. |
| ER | Thank you so much for your donation, it will help many children. |
| EE | Thank you. I hope you will continue to donate to the charity. |

Table 11: A conversation between a machine persuader (ER) and a machine persuadee (EE) trained by ARDM in self-play setting

| Role | Utterance |
| --- | --- |
| ER | Hi there! |
| EE | Hi, I'm Carolyn. How are you today? |
| ER | I am good, thank you. I am also here to talk about a charity called Save the Children. Do you have any idea about that? |
| EE | I think it's something that I should know about, but I don't have the funds. |
| ER | I understand that, I know that for sure. |
| EE | I am glad to hear that. How does it help children in the USA? |
| ER | It is really great. I am here to talk to you about something that is really important to me. I am here to tell you about something that I am really passionate about. |
| EE | I think it's a very important issue. I don't want to donate to charities that I know are not helping children. |
| ER | Yes, it is. |
| EE | Do you donate regularly to others? |
| ER | I have not yet, I am not sure. |
| EE | I feel a little guilty for not donating when I think about the children I have lost. |
| ER | Yeah, I totally understand. |
| EE | I would like to donate to a charity that I feel would be able to help all the children in the world. |
| ER | Awesome! That is great. How much would you like to donate to Save the Children? |
| EE | I would like to donate $1.00 |
| ER | Great! That is awesome. |
| EE | I am happy to hear that. I have heard about Save the Children, and it is something that I really, truly need. |
| ER | That is so great of you. |
| EE | Thank you for the opportunity. |

Table 12: A conversation between a machine persuader (ER) and a machine persuadee (EE) trained by ARDM in self-play setting

| Role | Utterance |
| --- | --- |
| ER | hello how are you? |
| EE | I'm doing well, how are you? |
| ER | I'm doing well, thanks for asking. I was wondering if you would be willing to donate a portion of your task payment to save the children. |
| EE | I'm interested, how much will you donate? |
| ER | I would like to donate the entire amount, but I'm not sure how much I'd be able to donate. Do you have any other questions? |
| EE | I think I know the amount I would donate. |
| ER | I completely understand, I am hoping you would be willing to donate even a small amount. Save the children is an amazing charity that helps children in need around the world. |
| EE | How much do you think I can donate? |
| ER | I would like to donate a minimum of $2. I was wondering if you would be willing to donate any more, would you be willing to donate even a little bit of your task payment? |
| EE | I think I can donate $1. |
| ER | That's very kind of you. I would be happy to do that, but I can't think of any other charities I would be able to donate to. I am glad to hear that. |
| EE | I think you are very generous, and I am sure this will help in the future. |
| ER | It is wonderful to hear. Thank you for your donation. |
| EE | You are very welcome, and thank you for the opportunity to help. |

Table 13: A conversation between a machine persuader (ER) and a machine persuadee (EE) trained by ARDM in self-play setting

| Role | Utterance |
|------|-----------|
| ER | Hi, how are you today? |
| EE | I'm doing well! How are you? |
| ER | I am doing well as well, I hope you have a nice day. |
| EE | I am as well. |
| ER | Do you have children? |
| EE | Yes, I do. I have three boys and one girl. |
| ER | Have you ever donated to a charity? |
| EE | No. I don't. |
| ER | Have you heard of Save the Children? |
| EE | I have not. |
| ER | The charity is an international non-governmental organization that promotes children's rights, provides relief and helps support children in developing countries. |
| EE | I think they are a good organization. |
| ER | I am sure they would be happy to help. |
| EE | I think they would. |
| ER | Do you donate to charities? |
| EE | I do. |
| ER | How much do you like to donate to the charity now? Your donation will be directly deducted from your task payment. You can choose any amount from $0 to all your payment ($2). |
| EE | I would like to donate $0.50. |
| ER | That is very generous of you. |
| EE | I hope they are able to help a lot of children. |

Table 14: A conversation between a machine persuader (ER) and a machine persuadee (EE) trained by ARDM in self-play setting

