# OpenReview forum: "Alternating Recurrent Dialog Model with Large-Scale Pre-Trained Language Models"
_ICLR.cc/2020/Conference — Reject_

### Official Review · AnonReviewer3 · 2019-10-19
**Official Blind Review #3**

**Rating:** 1

**Review:**

This paper explores methods to incorporate a pretrained language model into a dialog system. The authors propose Alternating Recurrent Dialog Model (ARDM) where the pretrained language model is used to initialize both the user LM and the system LM. The authors also present a memory module to augment this dialog system where each memory slot contains a <key, value> pair derived from hidden states for each dialog turn.

I think the writing of the model section could be improved.
A few clarification questions for the authors:
- How is the hidden state for the dialog turn t (i.e., h_t) derived from hidden states of tokens in turn t? Supposed there are N tokens in turn t, which Transformer hidden state is used as h_t?
- How is h_t used to compute p(w_i | w_{<i}, u_{<t}, s_{<t})?
If I understand correctly, the authors use t to index dialog turns, but then they seem to use the same symbol to denote token indices in the same section. So the exact model, although appears to be simple, is very confusing to me.

In terms of experiment results, the authors show that their approach improves over the basic GPT-2 and is competitive with baseline methods that rely on more supervision.A few clarification questions regarding the experiments:
- Did the authors tune the GPT-2 model on each dataset, similar to ARDM as well? Or are the GPT-2 results shown in Table 1 after fine-tuning?

In summary, this paper is a plug-and-play extension of the GPT-2 pretrained model that could be explained more clearly.

**Experience Assessment:**

I have published in this field for several years.

**Review Assessment: Checking Correctness Of Derivations And Theory:**

I assessed the sensibility of the derivations and theory.

**Review Assessment: Checking Correctness Of Experiments:**

I assessed the sensibility of the experiments.

**Review Assessment: Thoroughness In Paper Reading:**

I read the paper thoroughly.

---

> ### Author Response · Authors · 2019-11-07
> **Thank you for your time in reviewing our paper, and sorry for the confusion.**
>
> Sorry for the confusion, t is used to index dialog turns. We will use n to index tokens for clarity. So at the dialog turn t, h_t means all the hidden states for all N tokens, and it consists of h_t_1:h_t_N.
>
> Also, GPT-2 results in Table 1 are after fine-tuning, not zero-shot. Since it is not working well in CamRest676 to capture the speaker role information, we didn’t report its results on other datasets.
>
> Finally, we want to mention that our model modifies the original GPT-2 model to better adapt to dialogs by modeling different speakers separately. Our work can be helpful in exploring how to effectively use the large scale pre-trained language models for dialogs. This direction is getting more attention and has become an important research topic to improve the current dialog systems.

---

### Official Review · AnonReviewer1 · 2019-10-20
**Official Blind Review #1**

**Rating:** 3

**Review:**

The paper is concerned with reducing the amount of manual dialog state or dialog act labeling in task-oriented dialog, following the lead of Eric and Manning (2017) who did not require any such explicit annotation. The authors note that Eric and Manning’s approach didn’t do well on more recent dialog datasets in comparison to e.g. Sequicity (Lei et al., 2018), which trains an explicit belief state. The first main contribution of the submission is to exploit GPT-2 to outperform Sequicitiy and other techniques, doing so without explicit dialog state/act supervision. The second contribution is an alternating parameterization of the model that distinguishes between agent and user utterances. Experimental results show that the method of the paper (ARDM) is either superior or on par with methods that exploit hand-labeled dialog states/acts.

Originality/impact:

I think the approach of the paper is well motivated, though quite simple. The idea of improving on a generation task with GPT-2 is a tried one, and its effectiveness isn’t really surprising. The alternating parameterization is interesting but has been done before in (e.g.) Zhao and Kawahara (2019) and (Zhang et al., 2019) [1]. Overall, I think the paper doesn’t bring much novelty, but results are promising, and this work represents a step towards making task-oriented dialog system less reliant and hand coded linguistic information. It could have an impact for the dialog community.

Empirical contribution:

My main concern with the paper is a lack of ablation, which makes it hard to understand where the improvement comes from. More specifically:

* It is not known how much the alternating nature of ARDM contributes the overall gains compared to (1) other details of the model; (2) pre-training with GPT-2. One way to ablate ARDM’s alternating mechanism from other characteristics of the model would be for example to tie the parameters of the user and the agent.

* Table 1 compares GPT-2 with ARMD, so we don’t know if the improvement comes from either fine tuning on CamRest676, from ARDM’s alternation between speakers, or from other characteristics of ARDM. Note: The text characterizes GPT-2 as “a pre-trained large-scale language model GPT-2”, so I presume “GPT-2” here means without fine-tuning.

* Results of Table 3 are perplexing and more analyzes would be needed to understand what is really happening. Considering that both TransferTransfo and ARDM are based on transformer and GPT-2, and are apparently fine-tuned on the same data, I find it strange the gap between the two is so big on perplexity (10.1 vs TransferTransfo’s 19.9) while TransferTransfo is actually superior on BLEU. I understand BLEU is problematic for dialog evaluation as it treats references (gold responses) as the only reasonable responses (Liu et al., 2016), but that makes perplexity as an evaluation metric problematic for exactly the same reason. In fact, (Schwenk et al., 2006)[2], (Luong et al., 2016)[3], and others have shown that BLEU and perplexity are highly correlated, which isn’t surprising as both penalize generated responses to the extent that they lexically differ from the gold standard. So, these results look a bit strange/suspicious to me, and I think deserve further investigation. Sample outputs of both TransferTransfo and ARDM might help better understand the problem (Why showing only ARDM’s? When showing generation examples, I think it is standard practice to show outputs of multiple systems, and not just for the system of the paper).

Minor comments:

- [Manning and Eric, 2017] should be [Eric and Manning, 2017]
- ARDM with “recurrent” maybe isn’t a good name as it may imply it is RNN instead of Transformer based.
- Section 3.1: I suggest you explain the “memory mechanism” more formally and in more details, e.g., M_{t-1} doesn’t seem to be used anywhere.
- “assume there is only one layer in Transformer”: Is it for the sake of the presentation only, I assume?
- Tables 1-3: perhaps add confidence intervals as some of the differences are quite small, and so are some of the test sets.
- Table 3 doesn’t seem to be referenced anywhere.
- Table 4: I would recommend showing ARDM and TransferTransfo side by side.

[1]: https://arxiv.org/abs/1903.05759
[2]: https://www.aclweb.org/anthology/P06-2093/ (e.g., Fig. 5 shows the high correlation between BLEU and perplexity)
[3]: https://arxiv.org/abs/1410.8206

Questions:

(1) In 4.3.1, are capacities of the two models the same? If ARDM has (roughly) double the number of parameters due to its modeling of two speakers, what about comparing TransferTransfo at same model capacity to make the comparison more meaningful?

(2) I am confused how you (the authors) compute belief state success F1 for GPT-2, as the text says there is only pre-training with this model, which I think implies there is no fine-tuning on the belief state identification task. Could you provide more details?

**Experience Assessment:**

I have published in this field for several years.

**Review Assessment: Checking Correctness Of Derivations And Theory:**

I carefully checked the derivations and theory.

**Review Assessment: Checking Correctness Of Experiments:**

I carefully checked the experiments.

**Review Assessment: Thoroughness In Paper Reading:**

I read the paper at least twice and used my best judgement in assessing the paper.

---

> ### Author Response · Authors · 2019-11-07
> **Thanks for your reviews.**
>
> First, thank you for the reviews. Sorry for the confusion on the name GPT-2 in Table 1. We will update the entry named, GPT-2  in Table 1 to GPT-2-Finetune. Here, it stands for the GPT-2 fine-tuned on CamRest676. Without fine-tuning on the targeted dataset, vanilla GPT-2 can’t even generate the requested name slot information. So we did not have it as the baseline in Table 1. You can also think GPT-2-Finetune is our model, ARDM without the alternating mechanism,  just uses one language model to learn the entire dialog.
>
> For the second question, TransferTransfo is only slightly better than ARDM’s in terms of BLEU score (17.0 vs 16.5). So it is possible in terms of perplexity, ARDM can better than TransferTransfo. Also a random sentence with common tokens “the, of, is, are…” already has 10.0+ BLEU-1 score. We would not refer readers to take much trust in BLUE score.  Another reason under the uncorrelation between BLEU and perplexity is that there are only 100 dialogs in validation set (we only have 1,017 dialogs in total). The number of dialogs are limited so there might be variance in the experiments we cannot account for even we did randomly selected the 100 dialogs for validation. In sum, the automatic scores for PersuasionForGood is not really very meaningful. The meaning scores are the human evaluation where we randomly assign the two models for humans to chat with. We collected 20*20 = 400 dialogs in human evaluation.  Human evaluation reflects more the model quality better. We also published our code, sample Colab examples and generated dialogs in here to provide more evidence. In the paper, we will update the result discussion and include TransferTransfo generated dialogs to provide users with a more comprehensive view of each model’s performance.
>
> For the question (1): ARDM doubled the number of parameters compared to TransferTransfo. For fairness, we will conduct an experiment to expand TransferTransfo with the same amount to parameter.
> However, we want to also stress the advantage of our model, ARDM. Although, it has more parameters, it has the same time and memory complexity compared to TransferTransfo. The two models’ computation have the same number of floating operations.  In the future, we will analyze how bigger models affect generation performance by using GPT-2 medium and large.
>
> For the question (2): Sorry for the confusion. The GPT-2 here is finetuned on the target task-oriented dialogue. We should use a different name, such as GPT-2-finetune to indicate that. Indeed, without finetune, the system can’t generate requested slots.

---

### Official Review · AnonReviewer2 · 2019-10-23
**Official Blind Review #2**

**Rating:** 8

**Review:**

This paper proposes a pre-trained language model architecture specifically used for task-oriented dialogue systems. The basic idea is to alternate between the likelihood of two parties in a dialogue. This simple yet effective approach yield significant improvements in baseline dialogue system datasets in terms of both BLEU score and accuracies. Experiments are done throughly by comparing to BERT and GPT-2, which reflected the cutting edge research in pre-trained language model. This paper opened up potential new domains that can inspire a wide range of work and it fits very well into the ICLR community.

**Experience Assessment:**

I have published in this field for several years.

**Review Assessment: Checking Correctness Of Derivations And Theory:**

I assessed the sensibility of the derivations and theory.

**Review Assessment: Checking Correctness Of Experiments:**

I carefully checked the experiments.

**Review Assessment: Thoroughness In Paper Reading:**

I read the paper thoroughly.

---

> ### Author Response · Authors · 2019-11-07
> **Thanks for the comments.**
>
> Thank you for reviews. We will continue to explore this direction of using a general pre-trained dialog model to improve different dialog tasks.

---

### Author Response · Authors · 2019-11-08
**For whoever wants to play with it, we provide a rough demo of implementation with pre-trained weights**

https://colab.research.google.com/drive/1cd-jTWqMnQST4vmnz1Ygb0VDkqu_jXDZ

---

### Decision · Program_Chairs · 2019-12-19

**Decision:**

Reject

**Comment:**

This paper proposes an alternating dialog model based on transformers and GPT-2, that model each conversation side separately and aim to eliminate human supervision. Results on two dialog corpora are either better than or comparable to state-of-the-art. Two of the reviewers raise concerns about the novel contributions of the paper, and did not change their scores after authors' rebuttal. Furthermore, one reviewer raises concerns about the lack of detailed experiments aiming to explain where the improvements come from. Hence,  I suggest rejecting the paper.